# In vivo MRI is sensitive to remyelination in a nonhuman primate model of multiple sclerosis

**Maxime Donadieu[1†], Nathanael J Lee[1,2†], María I Gaitán[1], Seung-Kwon Ha[1,3], Nicholas J Luciano[1], Snehashis Roy[4], Benjamin Ineichen[1,5], Emily C Leibovitch[6], Cecil C Yen[7], Dzung L Pham[8], Afonso C Silva[3,7], Mac Johnson[9], Steve Jacobson[6], Pascal Sati[1,10]\*, Daniel S Reich[1]\***

[1]Translational Neuroradiology Section, Laboratory of Functional and Molecular Imaging, National Institute of Neurological Disorders and Stroke, National Institutes of Health, Bethesda, United States; [2]Department of Neurology and Neurological Sciences, Stanford University, Palo Alto, United States; [3]Department of Neurobiology, University of Pittsburgh, Pittsburgh, United States; [4]Section on Neural Function, National Institute of Mental Health, National Institutes of Health, Bethesda, United States; [5]Department of Neuroradiology, Clinical Neuroscience Center, University Hospital Zurich, University of Zurich, Switzerland, Switzerland; [6]Viral Immunology Section, Laboratory of Functional and Molecular Imaging, National Institute of Neurological Disorders and Stroke, National Institutes of Health, Bethesda, United States; [7]Cerebral Microcirculation Section, Laboratory of Functional and Molecular Imaging, National Institute of Neurological Disorders and Stroke, National Institutes of Health, Bethesda, United States; [8]Department of Radiology and Radiological Sciences, Uniformed Services University of the Health Sciences, Bethesda, United States; [9]Vertex Pharmaceuticals Incorporated, Boston, United States; [10]Neuroimaging Program, Department of Neurology, Cedars Sinai, Los Angeles, United States

**\*For correspondence:**
pascal.Sati@cshs.org (PS);
daniel.reich@nih.gov (DSR)

[†]These authors contributed equally to this work

**Abstract** Remyelination is crucial to recover from inflammatory demyelination in multiple sclerosis (MS). Investigating remyelination in vivo using magnetic resonance imaging (MRI) is difficult in MS, where collecting serial short-interval scans is challenging. Using experimental autoimmune encephalomyelitis (EAE) in common marmosets, a model of MS that recapitulates focal cerebral inflammatory demyelinating lesions, we investigated whether MRI is sensitive to, and can characterize, remyelination. In six animals followed with multisequence 7 T MRI, 31 focal lesions, predicted to be demyelinated or remyelinated based on signal intensity on proton density-weighted images, were subsequently assessed with histopathology. Remyelination occurred in four of six marmosets and 45% of lesions. Radiological-pathological comparison showed that MRI had high statistical sensitivity (100%) and specificity (90%) for detecting remyelination. This study demonstrates the prevalence of spontaneous remyelination in marmoset EAE and the ability of in vivo MRI to detect it, with implications for preclinical testing of pro-remyelinating agents.

## Editor's evaluation

This important study combined MRI(magnetic resonance imaging) imaging and histopathology to examine the remyelination of brain lesions in an EAE marmoset model of multiple sclerosis. This work addresses in a non-human primate a missing link in the neuropathology of myelin repair, because in human MS it is virtually impossible to study the lesion dynamics by MRI (in live patients)

and demyelination by histology (upon brain autopsy). The data presented are solid although the conclusions would have been stronger with further histological evidence of remyelination.

## Introduction

Multiple sclerosis (MS) is a debilitating inflammatory demyelinating disorder affecting millions worldwide (*Reich et al., 2018*). MS causes dynamic changes to myelin in the central nervous system, including the quintessential focal inflammatory destruction of myelin, as well as the phenomenon of remyelination that can follow the demyelination (*Lubetzki et al., 2020*; *Brown et al., 2014*; *Chari, 2007*; *Lassmann et al., 1997*). Demyelinated axons are susceptible to persistent damage and neurological dysfunction in MS; therefore, remyelination is a crucial aspect of tissue repair, and as such represents an important therapeutic target (*Kremer et al., 2019*). Most knowledge about remyelination in MS derives from postmortem studies using histochemistry and electron microscopy. This is because investigating remyelination in vivo in real time is limited by imperfect discrimination on neuroimaging modalities such as magnetic resonance imaging (MRI). Furthermore, in human beings, where collecting serial short-interval scans is challenging, it is difficult to detect and track the dynamic occurrence of remyelination. Therefore, to investigate the pathobiology of remyelination in the context of focal inflammatory demyelination, a reliable preclinical model is needed to develop techniques that can then be applied clinically.

Rodent models have been widely used to investigate various aspects of the pathobiology of demyelination. However, while toxin models in mice, including the lysolecithin and cuprizone models, display demyelination and even remyelination, they do not recapitulate the recruitment of peripherally derived adaptive immune cells that occurs in MS, which can potentially confound the MRI signal (*Kipp et al., 2012*; *Kipp et al., 2017*). Conversely, rodent experimental autoimmune encephalomyelitis (EAE) models, though driven by an immune response, are often neither focally nor profoundly demyelinating and mostly impair the spinal cord. There is no known rodent model that is characterized by multifocal inflammatory demyelination in the brain that resembles MS and is disseminated in both space and time.

However, EAE in the common marmoset (*Callithrix jacchus*) is a well-recognized translational model that serves as a bridge between the rodent EAE and human MS (*Jagessar et al., 2016*; *Kap et al., 2010*; *Kap et al., 2016*). Not only does EAE resemble MS white matter lesions (WML) both radiologically and pathologically (*Sati et al., 2012*; *Maggi et al., 2017*), but marmoset WML spontaneously remyelinate (*Lee et al., 2019*), as occurs in MS.

Prior studies demonstrated that certain signal changes in MRI, such as magnetization transfer ratio (MTR), correlate with demyelination and remyelination in MS lesions (*Chen et al., 2008*; *Chen et al., 2013*; *Filippi et al., 2012*; *Absinta et al., 2016*; *Laule and Moore, 2018*). This has also been investigated in animal models, albeit mainly in rodent cortex (*Yano et al., 2018*; *Schmierer et al., 2010*) rather than white matter. It has also been demonstrated that partial remyelination can occur and can be localized either to specific parts of the lesion (most commonly the lesion edge) or to the whole lesion (*Filippi et al., 2019*; *Patrikios et al., 2006*).

Here, we studied focal WML in marmoset EAE. We utilized serial in vivo MRI, mainly involving proton density-weighted (PDw), T1-weighted (T1w) before and after intravenous injection of a gadolinium chelate, and MTR sequences, to age and characterize lesions and predict remyelination. We further analyzed WML histopathology, focusing on myelin lipids and proteins as well as mature oligodendrocytes and their precursors, to compare and study the reliability of using various in vivo MRI sequences to predict remyelination in the context of recovery from inflammatory demyelination.

## Results

### Lesion characterization and categorization on MRI

EAE was induced in six marmosets as described, and the animals followed by MRI at regular intervals until sacrifice due to clinical progression. Lesions were grouped into three categories. Those categorized as 'early active' at histopathology typically remained hyperintense on PDw images until the terminal scan and grew rapidly to several cubic millimeters, subsequently showing minimal lesion volume change over time (example in *Figure 1A–B*) and were enhancing on terminal T1w MRI with

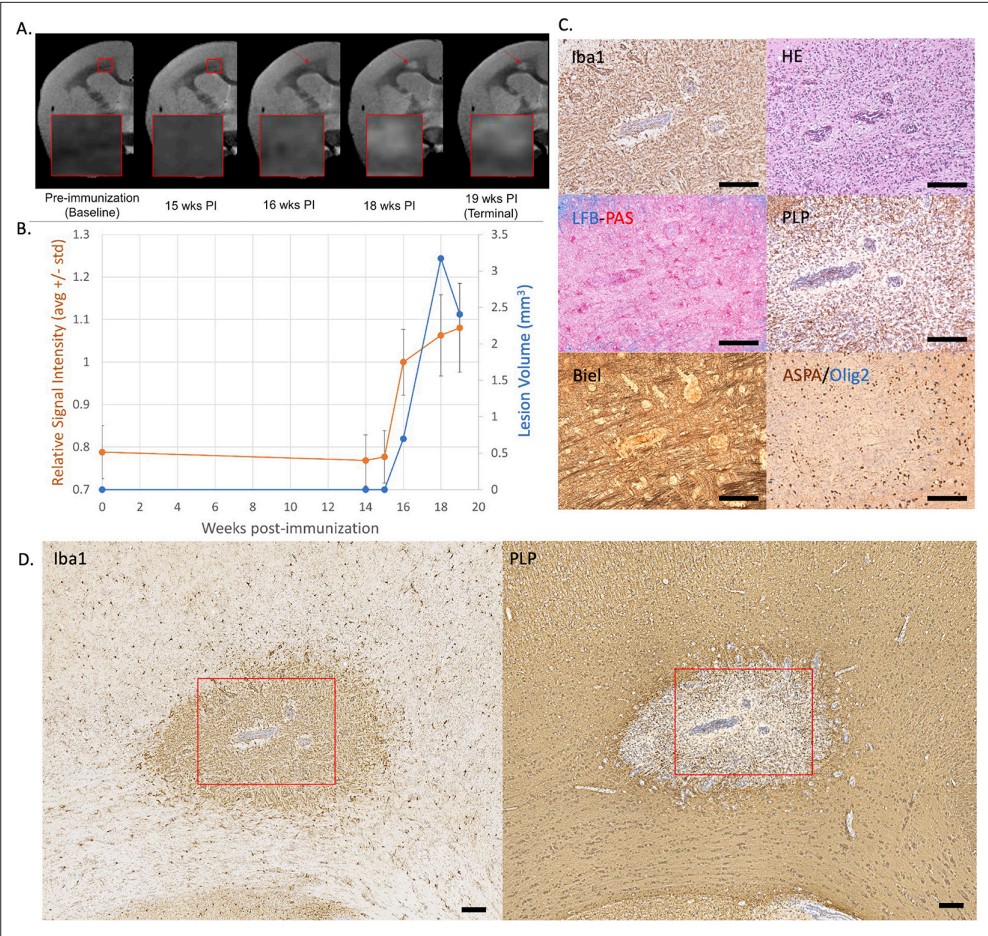

**Figure 1.** Example of an early active demyelinating EAE lesion with persistent hyperintensity on PDw MRI and active inflammatory demyelination on histopathology. (**A**) In vivo PDw MRI acquired before EAE induction (baseline) and at various timepoints leading up to the terminal scan. Images were processed as described in Methods. Red arrows: focal white matter lesion first detected 16 weeks after immunization, which persisted through the terminal MRI 3 weeks later. Red boxes: location of magnified insets. (**B**) Temporal evolution of volume (blue line) and normalized PDw signal intensity (orange line) of the segmented lesion. (**C**) Histochemical panel magnification of the same lesion, demonstrating inflammation (Iba1+microglia/macrophage infiltration) and demyelination (loss of normal PLP staining). (**D**) Higher magnification images from the center of the lesion (red boxes in C) showing increased cellularity, loss of myelin lipid (LFB), partial loss of oligodendrocytes and their precursors (ASPA/Olig2), partial loss of axons (Biel), and edema (increased intercellular spaces). Scale bars = 200 μm. Hematoxylin counterstaining used for PLP and Iba1. Lesion selected from M#6. *Abbreviations:* EAE, experimental autoimmune encephalomyelitis; PDw, proton density-weighted; MRI, magnetic resonance imaging; PLP, proteolipid protein; LFB-PAS, Luxol fast blue–periodic acid Schiff; Biel, Bielschowsky; ASPA, aspartoacylase.

gadolinium contrast. Lesions classified as 'chronic, at least partially demyelinated,' demonstrated areas of PDw hyperintensity that persisted until the terminal scan despite resolution of early gadolinium enhancement on T1w images (*Figure 2A–B*). Lesions classified as 'remyelinated' were initially hyperintense on PDw images, with subsequent return toward isointensity over time (*Figure 3A–B*). None of the lesions returning to isointensity on PDw images demonstrated gadolinium enhancement on T1w images at terminal MRI.

## In vivo PDw MRI is sensitive to lesion remyelination

Using in vivo MRI only, 40 focal WML were detected in the 6 EAE marmosets (*Table 1*). Interrater reliability for PDw MRI classification was 94%, with Cohen's kappa of 0.89, and consensus was achieved across the raters for all lesions. Of the 40 lesions, 12 (30%) were classified as predicted early active, 10 (25%) as predicted chronic at least partially demyelinated, and 18 (45%) as predicted remyelinated.

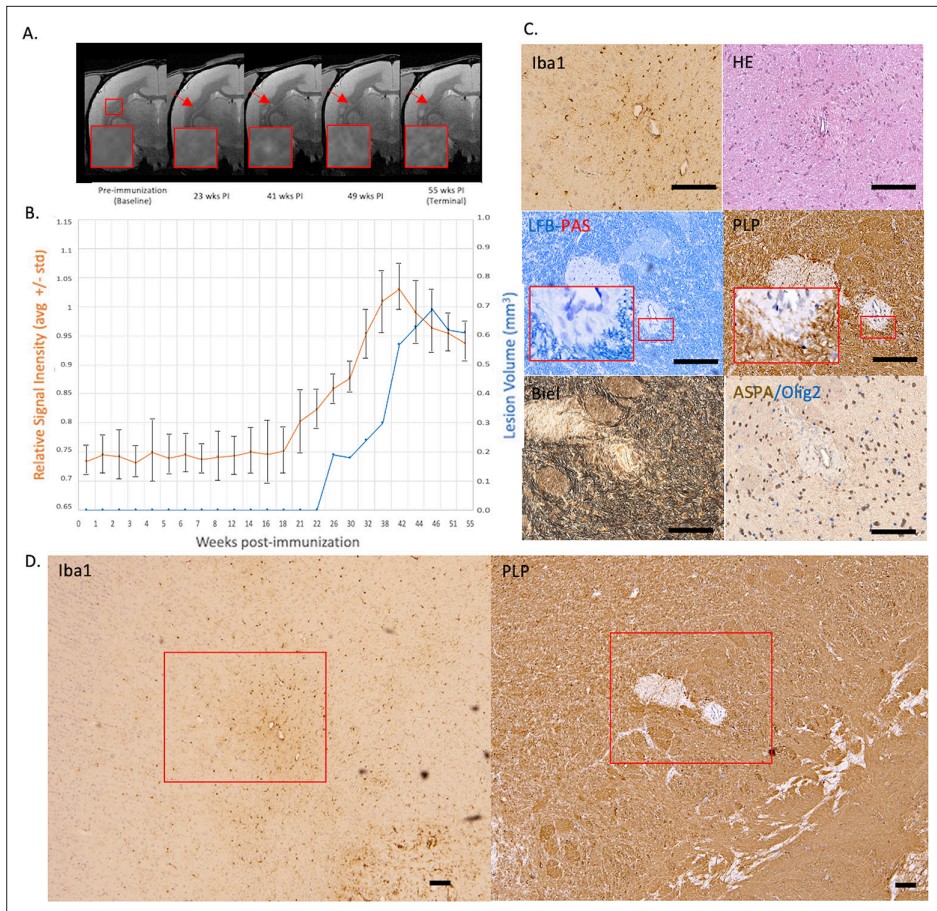

**Figure 2.** Example of a chronic, at least partially demyelinated EAE lesion core with long-lasting hyperintensity on serial PDw MRI and complete loss of myelin on histopathology. (**A**) In vivo PDw MRI acquired before EAE induction (baseline) and at various timepoints leading up to the terminal scan. Images were processed as described in Methods. Red arrows: focal white matter lesion first detected 23 weeks after immunization, which persisted through the terminal MRI 32 weeks later. Red boxes: location of magnified insets. (**B**) Temporal evolution of volume (blue line) and normalized PDw signal intensity (orange line) of the segmented lesion. (**C**) Histochemical panel magnification of the same lesion, demonstrating mild inflammation (Iba1+ microglia/macrophage infiltration) as well as demyelination (loss of normal PLP staining). (**D**) Higher magnification images from the center of the lesion (red boxes in C) showing minimally increased cellularity, partial loss of myelin lipid (LFB), loss of oligodendrocytes and their precursors (ASPA/Olig2), and loss of axons (Biel). Scale bars = 200 μm. Hematoxylin counterstaining used for PLP and Iba1. Lesion selected from M#3. *Abbreviations:* EAE, experimental autoimmune encephalomyelitis; PDw, proton density-weighted; MRI, magnetic resonance imaging; PLP, proteolipid protein; LFB-PAS, Luxol fast blue–periodic acid Schiff; Biel, Bielschowsky; ASPA, aspartoacylase.

Four of the six animals demonstrated predicted remyelinated lesions (M#1–4), whereas the remaining two only had predicted acute demyelinating and chronic, at least partially demyelinated lesions (M#5–6).

Based on the histological classification criteria, 31 lesions were identified in the 5 animals with postmortem tissue (M#1 was not included in the histological analysis). Twelve of the 31 lesions (39%) were classified as early active, 10 (32%) as chronic at least partially demyelinated, and 9 (29%) as remyelinated (*Table 2*). All three animals with predicted remyelinated lesions on MRI (M#2–4) had remyelinated lesions on histology. M#5 and M#6 only harbored early active and chronic, at least partially demyelinated lesions, consistent with MRI findings. Two lesions in M#2 and one in M#4 were identified as predicted remyelinated on MRI but chronic, at least partially demyelinated on histology.

Of the 31 focal WML identified on both in vivo PDw MRI and histology, classification was concordant for 27 WML (87%). When lesions were grouped by myelination status only (i.e. early active or chronic, at least partially demyelinated vs. remyelinated), in vivo PDw MRI predicted 19 demyelinated

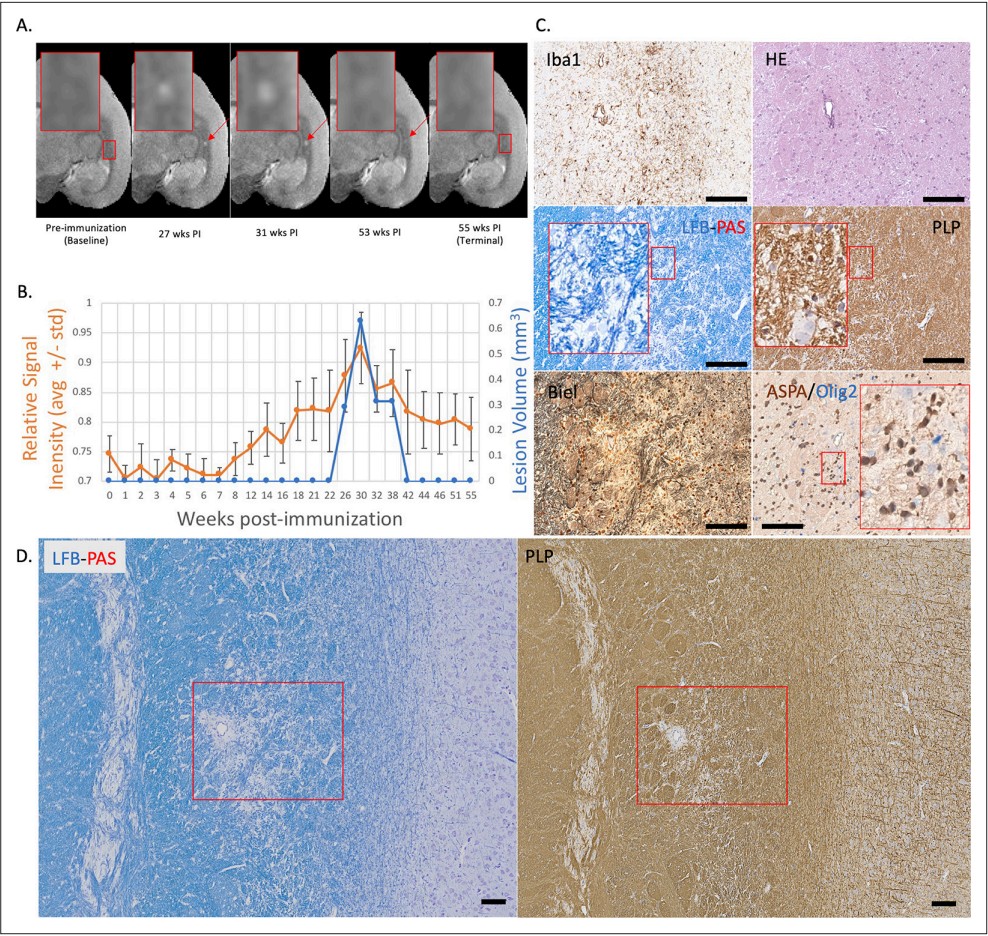

**Figure 3.** Example of a nearly complete remyelinated EAE lesion with initial hyperintensity that returned to isointensity on serial PDw MRI. (**A**) In vivo PDw MRI acquired before EAE induction (baseline) and at various timepoints leading up to the terminal scan. Images were processed as described in Methods. Red arrows: focal white matter lesion first detected 27 weeks after immunization, which largely resolved on MRI and could not be reliably segmented on the terminal MRI 32 weeks later. Red boxes: location of magnified insets. (**B**) Temporal evolution of volume (blue line) and normalized PDw signal intensity (orange line) of the segmented lesion. (**C**) Histochemical panel magnification of the same lesion, demonstrating pale myelin lipid staining (LFB) and near-normal levels of myelin protein (PLP). (**D**) Higher magnification images from the center of the lesion (red boxes in C) showing minimal inflammation (Iba1), presence of oligodendrocytes and their precursors (ASPA/Olig2), and partial preservation of axons (Biel). Scale bars = 200 μm. Hematoxylin counterstaining used for PLP and Iba1. Lesion selected from M#3. *Abbreviations:* EAE, experimental autoimmune encephalomyelitis; PDw, proton density-weighted; MRI, magnetic resonance imaging; PLP, proteolipid protein; LFB-PAS, Luxol fast blue–periodic acid Schiff; Biel, Bielschowsky; ASPA, aspartoacylase.

lesions and 12 remyelinated lesions, whereas histology showed 22 demyelinated and 9 remyelinated lesions (*Table 3*). Relative to histology, PDw MRI prediction was therefore 100% sensitive and 90% specific for remyelination.

## MTR is less sensitive to lesion remyelination than PDw

MTR was also used to classify lesions using a similar rater-based analysis to that applied to the PDw images. The sensitivity and specificity for predicting remyelination, relative to histology, were 82% and 79%, respectively.

## T1w gadolinium enhancement as a marker of acute inflammation

Across the six animals scanned longitudinally, 82% of the lesions newly detected on PDw MRI presented T1w gadolinium enhancement. Enhancement was occasionally seen at the following

**Table 1.** Classification of focal lesions by proton density-weighted magnetic resonance imaging.

| Animal | Predicted early active | Predicted chronic, at least partially demyelinated | Predicted remyelinated | Total |
|--------|------------------------|---------------------------------------------------|------------------------|-------|
| M#1 | 1 | 2 | 6 | 9 |
| M#2 | 1 | 1 | 6 | 8 |
| M#3 | 1 | 1 | 4 | 6 |
| M#4 | 0 | 1 | 2 | 3 |
| M#5 | 3 | 3 | 0 | 6 |
| M#6 | 6 | 2 | 0 | 8 |
| Total | 12 | 10 | 18 | 40 |

timepoint (10–15 days after first detection). No chronic, at least partially demyelinated or remyelinated WML presented gadolinium enhancement on the terminal scan. M#5–6 presented at least one early active WML enhancing lesion at their terminal scan.

## Lesion remyelination occurs over a 4- to 9-week period and is sensitive to lesion size

Based on longitudinal MRI imaging of EAE lesions and analysis of normalized PDw signal intensity, with comparison to histology, we found that inflammation and demyelination were dominant in lesions younger than 10 weeks of age, corroborating previous work (*Lee et al., 2019*; *Lee et al., 2018*). In four representative lesions that remyelinated, PDw signal intensity stabilized near baseline between 4 and 9 weeks after initial lesion detection on MRI (*Figure 4*). Lesions larger than 0.5 μL at peak, as measured on PDw MRI, did not return to isointensity and correspondingly remained at least partially demyelinated on histology. On the other hand, most lesions smaller than 0.5 μL returned to isointensity and appeared remyelinated on histology.

## Remyelination is independent of corticosteroid administration

Per protocol, three of the six marmosets were given corticosteroids for 5 consecutive days to determine whether this treatment might alter lesion fate (principally remyelination). However, we found no differences in the prevalence of predicted remyelinated lesions based on corticosteroid treatment status. On serial PDw MRI, 10 of 21 lesions (48%) were predicted to be remyelinated in corticosteroid-treated marmosets, compared to 8 of 19 (42%) in untreated animals. Based on histological analysis in animals with available tissue, 4 of 12 lesions (33%) in corticosteroid-treated marmosets were remyelinated, compared to 5 of 19 (27%) lesions in untreated marmosets. The point biserial correlation model analysis showed that steroid administration had no significant correlation with remyelination detected on MRI (p=0.8). The average experimental duration also did not differ (40 weeks in treated and 43 weeks in untreated marmosets; p=0.85). Histological analyses also did not reveal differences between treated and untreated lesions.

**Table 2.** Classification of focal lesions by histopathology.

| Animal | Early active | Chronic, at least partially demyelinated | Remyelinated | Total |
|--------|--------------|------------------------------------------|--------------|-------|
| M#2 | 1 | 3 | 4 | 8 |
| M#3 | 1 | 1 | 4 | 6 |
| M#4 | 0 | 2 | 1 | 3 |
| M#5 | 3 | 3 | 0 | 6 |
| M#6 | 7 | 1 | 0 | 8 |
| Total | 12 | 10 | 9 | 31 |

**Table 3.** Confusion matrix for PDw MRI prediction of demyelinated vs. remyelinated lesions compared to histology.

| n=31 | Predicted demyelination by MRI | Predicted remyelination by MRI | |
|---|---|---|---|
| Actual demyelination by histology | 19 | 3 | 22 |
| Actual remyelination by histology | 0 | 9 | 9 |
| | 19 | 12 | |

## Histological quantification recapitulates MRI rater analysis of lesion myelin status

Assessment of proteolipid protein (PLP) staining in 31 lesions and 10 normal appearing white matter (NAWM) areas (1500 µm² centered over the lesion core) demonstrated larger unstained areas in early active (58 ± 25%) and chronic, at least partially demyelinated (38 ± 25%) lesions compared to remyelinated lesions (4.5 ± 1.1%) or NAWM (2.6 ± 0.3%) (*Figure 5A*). LFB assessment showed similar results: 63 ± 26% unstained area in early active lesions, 43 ± 26% in chronic, at least partially demyelinated lesions, 15 ± 25% in remyelinated lesions, and 3.0 ± 0.4% in NAWM (*Figure 5B*). We observed a significantly smaller unstained PLP area in NAWM compared to remyelinated lesions (two-sample t-test, p<0.001). There were no apparent differences between PLP and LFB staining in the different

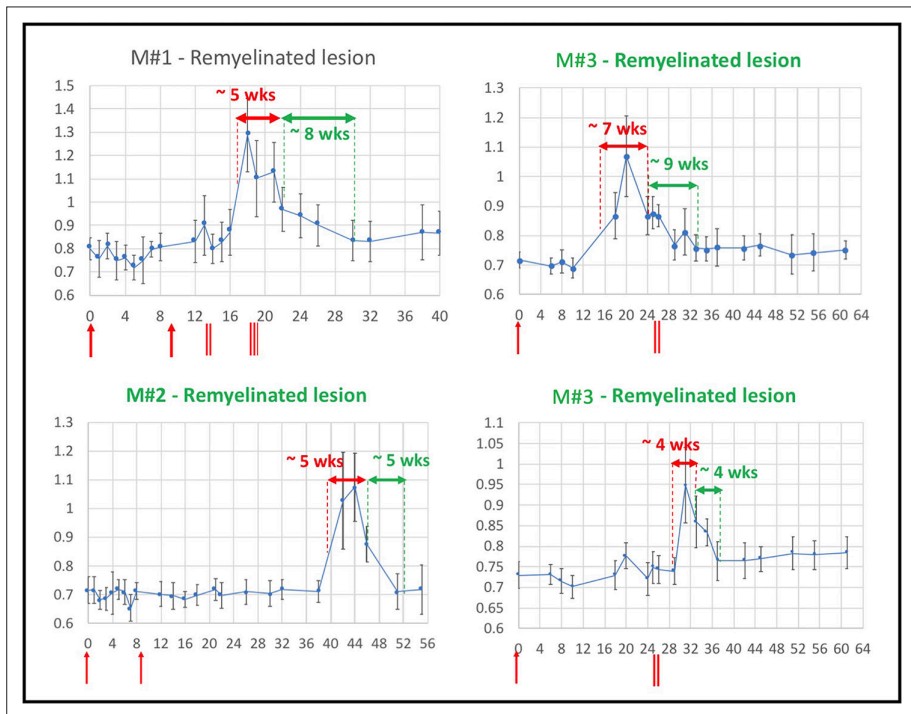

**Figure 4.** Evolution of in vivo PDw MRI signal intensity of remyelinating EAE lesions shows a typical 4- to 9-week time course of remyelination. Vertical axis: mean PDw signal intensity relative to gray matter. Horizontal axis: weeks post-immunization. Blue line corresponds to mean normalized PDw signal intensity (with standard deviation) of the segmented lesion, relative to gray matter signal intensity, quantified in a region of interest drawn manually and located in the normal appearing white matter area before the lesion appeared and kept constant over time. Normal white matter displays an average normalized signal intensity of 0.65–0.75. Vertical red arrows indicate EAE immunization. Vertical red bars indicate days when corticosteroid treatments were administered (M#1 was treated twice). Horizontal red double arrows indicate the estimated period of demyelination. Horizontal green double arrows indicate the estimated period of remyelination based on the downward slope of intensity measurement, followed by plateauing of signal intensity drop. Green titles indicate that the lesion subtype was confirmed with histopathology. M# corresponds to animal number in *Table 4*. *Abbreviations:* PDw, proton density-weighted; EAE, experimental autoimmune encephalomyelitis; MRI, magnetic resonance imaging; M, marmoset; wks, weeks.

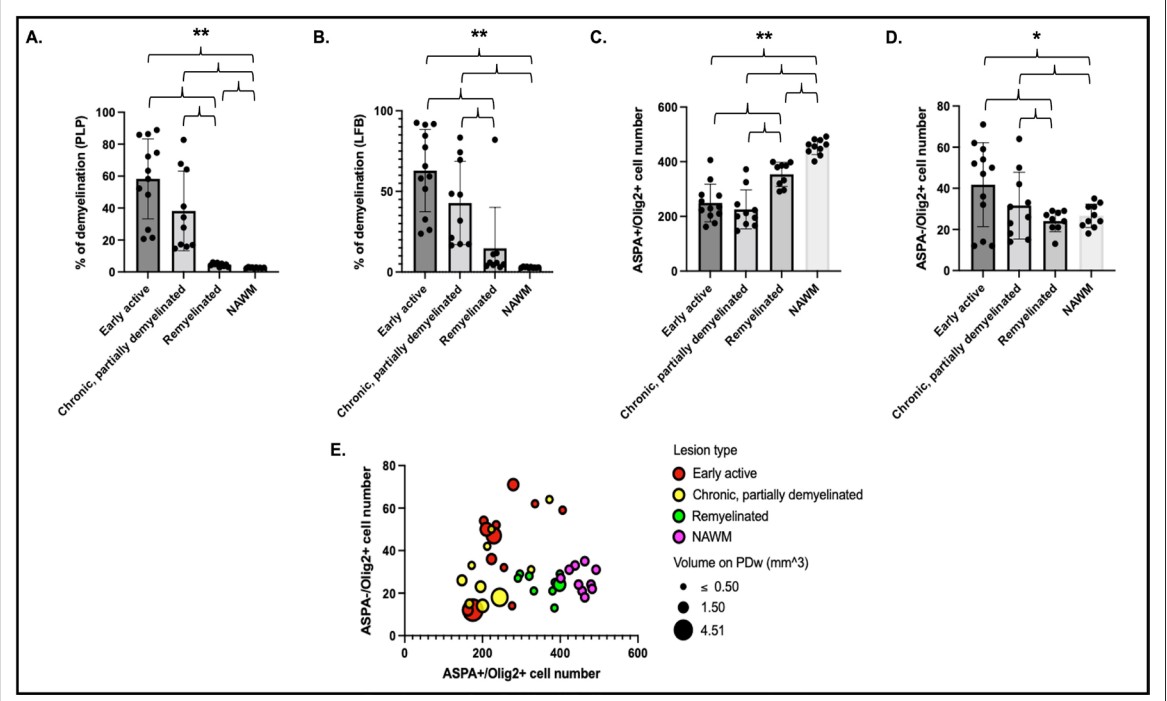

**Figure 5.** Histological quantification of myelin highlights different patterns across lesion categories and NAWM. Cells were quantified in a region of interest of 1500 μm² centered on each lesion core. Percentage of demyelination quantified for each lesion category and NAWM on PLP (**A**) and LFB (**B**). Number of ASPA and Olig2 double-positive oligodendrocytes (**C**) and ASPA negative, Olig2-positive OPC (**D**) in different lesion categories and NAWM. Data in A and B highlight more demyelination in acute and chronic lesions compared to remyelinated lesions or NAWM. Data in C and D highlight more mature oligodendrocytes in remyelinated lesions and NAWM compared to acute and chronic lesions, as well as more OPC in acute lesions compared to chronic lesions, remyelinated lesions, or NAWM. (**E**) Bubble plot of the 31 lesions and NAWM investigated with histopathology. Vertical axis: counts of ASPA- Olig2+ OPC. Horizontal axis: counts of ASPA+ Olig2+ oligodendrocytes. Bubble color: red; acute lesions, yellow; chronic lesions, green; remyelinated lesions, purple; NAWM. Bubble size: volume on PDw at the terminal scan in mm³. ANOVA: *p<0.05, **p<0.005. *Abbreviations:* NAWM, normal appearing white matter; PLP, proteolipid protein; LFB, Luxol fast blue; ASPA; aspartoacylase; ANOVA, analysis of variance; OPC, oligodendrocyte progenitor cells; PDw, proton density-weighted.

lesion categories and in NAWM (two-sample t-test). Interestingly, one lesion appeared remyelinated on PLP, with less than 6% of unstained area, but demyelinated on LFB (82% of unstained area).

## Oligodendrocyte and OPC counts are consistent with degree of demyelination in lesions

Quantitative assessment of ASPA+/Olig2+ (mature oligodendrocytes) and ASPA-/Olig2+ (oligodendrocyte precursor cell [OPC]) across the 31 lesions and 10 NAWM areas showed, as expected, more mature oligodendrocytes in remyelinated lesions and NAWM than early active or chronic, at least partially demyelinated lesions (*Figure 5C*). Consistent with PLP observations, we observed significantly more oligodendrocytes in NAWM compared to remyelinated lesions (two-sample t-test, p<0.001). Interestingly, more OPC were found in early active demyelinating lesions than in remyelinated lesions or NAWM (*Figure 5D*). Younger lesions (<10 weeks of age by MRI) had more OPC than older lesions, highlighted by a negative correlation between lesion age and OPC count (r=–0.46; p=0.009).

## Discussion

In this study, we determined whether high-resolution, serial, in vivo conventional PDw MRI can effectively predict remyelination status in focal WML of marmoset EAE, a relatively faithful MS model. We found that remyelination is relatively common in this model: four of the six marmosets studied had remyelinated lesions on either MRI or histopathology, and nearly half (18) of the 40 lesions identified on MRI were predicted to be remyelinated. In people, WML repair is an age-dependent process that is variable across patients and clinical disease classification (*Chari, 2007*; *Lassmann et al., 1997*;

*Patrikios et al., 2006*). Interestingly, the two animals that did not show remyelination by in vivo MRI, M#5 and M#6, had the shortest experimental duration (16–18 weeks, compared to 40–61 weeks for the other four animals), suggesting more aggressive lesions and/or insufficient time for remyelination.

Our characterization of remyelinated lesions showed low residual levels of inflammation (microglia/macrophages) and an extensive but incomplete (using NAWM as reference) repopulation of mature oligodendrocytes, specifically a higher number of ASPA+/Olig2+ cells in the lesion core compared to chronic, at least partially demyelinated lesions. Lower percentage of unstained area on LFB and PLP slides (<6%), consistent with data from MS (*Chari, 2007*; *Lucchinetti et al., 2000*; *Lucchinetti et al., 1999*; *Patani et al., 2007*), was also observed in remyelinated lesions. Remyelinated lesions were nearly isointense to healthy white matter at the terminal in vivo PDw MRI timepoint and showed no enhancement on post-gadolinium T1w images.

Interestingly, one lesion appeared remyelinated on PLP with less than 6% of unstained area but demyelinated on LFB (82% of unstained area) and hyperintense on PDw MRI. This result suggests potentially distinct timelines for reconstitution of myelin lipids vs. protein in the context of remyelination (*Chari, 2007*). As previously described, MRI seems to be more sensitive to content of myelin lipid than of myelin protein (*Leuze et al., 2017*).

As expected, assessment of early active and chronic, at least partially demyelinated lesions showed fewer mature oligodendrocytes in the lesion core and edge as well as a percentage of demyelinated area >20% on LFB and PLP staining. Both early active and chronic, at least partially demyelinated were hyperintense on terminal in vivo PDw MRI. Taken together, these data suggest that in marmoset EAE, lesions that initially present as hyperintense foci on PDw MRI and subsequently return toward isointensity over time are undergoing remyelination. Our data further suggest that signal intensity time courses, as well as lesion size, can be used to infer lesion status in the setting of early lesion development and recovery, despite the fact that PDw MRI is also known to be sensitive to tissue-level factors other than myelin.

The time course of marmoset EAE lesion development and repair is relatively stereotyped, as confirmed here. Previous work from our group has shown that once lesions form, the period of demyelination lasts around 6 weeks (*Lee et al., 2019*), consistent with data from this study (4–7 weeks; see *Figure 4*). The time course of marmoset remyelination is somewhat slower than toxin-induced demyelinating models in rodents, further raising potential relevance of the marmoset EAE model for preclinical testing of putative remyelination therapies in MS (*McMurran et al., 2019*). Our data also suggest that, during the lifespan of a lesion, if its size exceeds ~0.5 µL, the lesion will most likely not start the repair process and remain chronically demyelinated.

Of the 12 early active lesions, 7 showed an increased number of ASPA-/Olig2+ cells in the lesion vicinity (*Figure 5*). This pattern, never observed in remyelinated lesions or NAWM, suggests a possible proliferative OPC response, which may indicate attempted repair or potentially an inflammatory role of OPC in this setting (*Harrington et al., 2020*).

In this study, we compared PDw and MTR MRI, since MTR is widely used to assay remyelination in vivo in MS (*Chen et al., 2008*; *Chen et al., 2013*; *Mallik et al., 2014*), and found that in the context of marmoset EAE, PDw, in close connection with histopathology, appears to have higher sensitivity and specificity for remyelination status. MTR may be low in remyelinated lesions compared to healthy white matter because of the presence of incomplete or morphologically altered myelin sheaths (*Brown et al., 2014*; *Mallik et al., 2014*). Unlike MTR, which suffers from signal-to-noise reduction due to its calculation as a voxel-wise division of signal intensity measures, PDw signal intensity is directly measured by the MRI system, and thus PDw may prove more reliable for simply discriminating the presence or absence of remyelination and for characterizing its time course.

To determine whether corticosteroid treatment alters the course of lesion repair, half of the marmosets (for each twin pair, the one that showed lesions first) received 5-day treatment courses. We found no differences by corticosteroid treatment status in prevalence or time course of remyelination on either MRI or histological quantification. However, it remains possible that optimal timing of corticosteroid treatment might influence remyelination. For example, it is possible that early initiation of corticosteroid treatment (as soon as the first lesion was detected) in this study, resulting in treatment completion before lesions were even 1 week old. This might have been too early to influence lesion outcome, as the initial demyelination period typically lasts 4–7 weeks before remyelination ensues. It is also possible that reduction of inflammation via corticosteroid treatment could have hampered

more rapid remyelination by slowing clearance of myelin debris, which is a prerequisite for OPC recruitment (*Cunha et al., 2020*) and differentiation (*Gruchot et al., 2019*; *Rawji et al., 2016*).

The main limitation of the study is its small sample size. This limitation is partially compensated by our focus on individual lesions, rather than the number of animals. Another limitation is that different EAE immunization protocols were applied across marmosets, though we did not observe any difference in lesion evolution or outcome either radiologically or histologically. Notably, previous studies from our group with a similar variety of EAE induction methods have also not shown consistent differences in disease course or lesion pathobiology (*Lee et al., 2019*; *Lee et al., 2018*). Finally, we did not obtain ultrathin sections to quantify myelin thickness, as we were interested in performing a battery of stains on our tissue, and as lesions were found at a variety of orientations relative to axons, which would complicate such quantification.

In conclusion, in vivo longitudinal PDw MRI can effectively predict remyelination in the context of marmoset EAE, with high sensitivity and specificity relative to histology. Given strong similarities between marmoset EAE and MS with respect to lesion development and repair, these results suggest a paradigm for preclinical – and possibly early-phase clinical – studies to investigate putative remyelinating therapies.

## Methods
### Marmoset EAE induction

Six marmosets (three pairs of twins; four males and two females, ages 2–6 at baseline; same group used for two different studies) were included in the study (*Table 4*). As a pilot study, two marmosets (M#1–2) first received 100 mg of human white matter homogenate followed by an additional 200 mg after no lesions were detected by in vivo MRI 2 months after the initial injection (*Lee et al., 2019*; *Lee et al., 2018*). Following protocol revision to boost the chance of early disease induction, four additional marmosets (M#3–6) received 200 mg of temporal-lobe white matter homogenate collected at autopsy. All white matter homogenates were mixed with complete Freund's adjuvant (Difco Laboratories). Data from the two induction protocols were combined for this study, as prior marmoset EAE studies have not revealed lesion-level pathology differences (*'t Hart and Massacesi, 2009*), and marmosets are a scarce nonhuman primate animal resource for which data use should be maximized.

In each twin pair, per protocol, the first animal to develop a lesion, as detected by in vivo MRI, received intravenous methylprednisolone (18 mg/kg/day for 5 consecutive days), the goal of which was to reduce severity of inflammation, potentially allowing longer-term evaluation of the lesions, and to detect a potential effect of corticosteroids on lesion repair. The specific regimen was based on treatment of acute MS relapses (*Goodin, 2014*; *Sellebjerg et al., 2005*), scaled to the marmoset setting. Per protocol, experiments were terminated when animals either became paraplegic or lost more than 20% of their baseline body weight.

**Table 4.** Demographics and experimental information for the six marmosets included in this study. Immunizations used human white matter homogenate. Experiment duration corresponds to the time between immunization and terminal MRI.

| Animal | Sex | Age (years) | First immunization | Second immunization | Corticosteroid treatment | Experiment duration (weeks) |
|---|---|---|---|---|---|---|
| M#1* | Male | 3.7 | 100 mg | 200 mg | Yes | 40 |
| M#2* | Male | 3.7 | 100 mg | 200 mg | No | 54 |
| M#3† | Male | 6.2 | 200 mg | – | Yes | 61 |
| M#4† | Male | 6.2 | 200 mg | – | No | 60 |
| M#5‡ | Female | 2.8 | 200 mg | – | Yes | 18 |
| M#6‡ | Female | 2.8 | 200 mg | – | No | 16 |

Denote the three different pairs of twin animals.

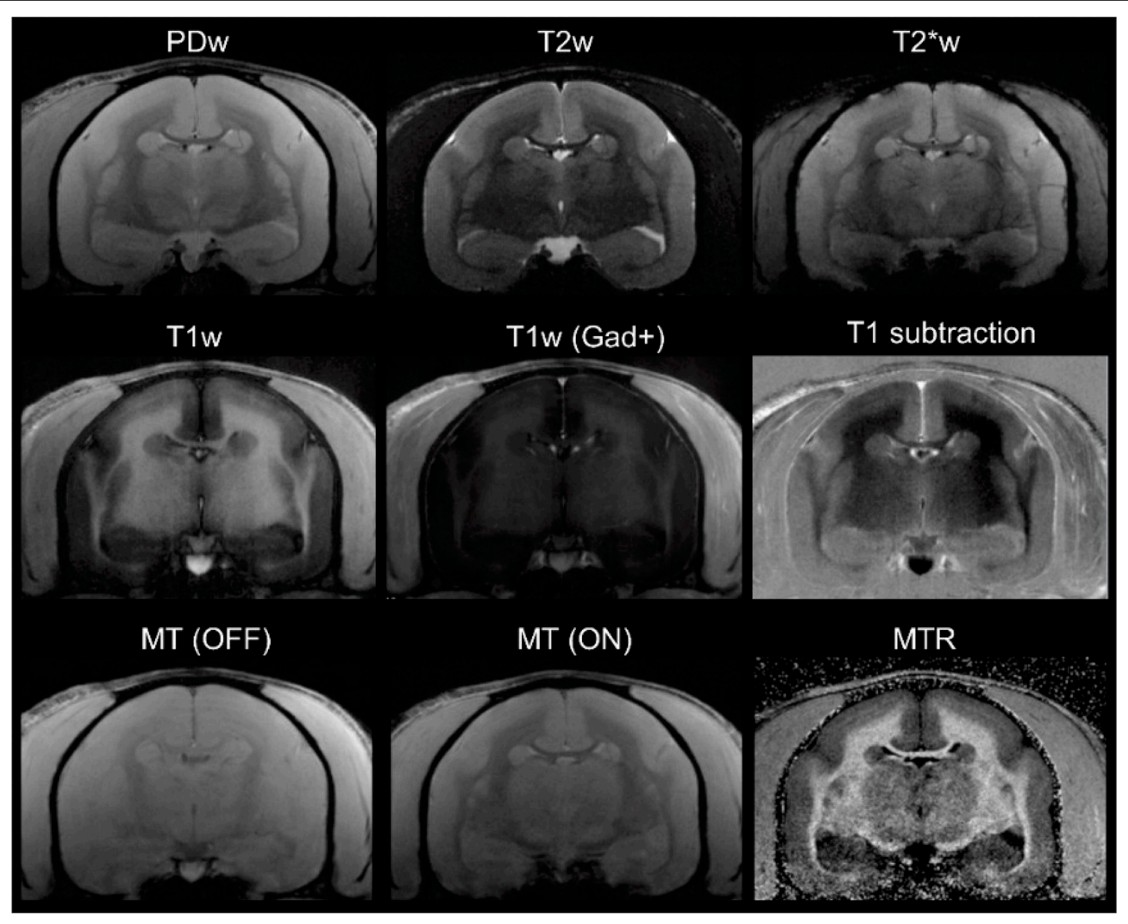

**Figure 6.** In vivo (baseline) multicontrast marmoset brain magnetic resonance imaging (MRI). The T1 subtraction image was obtained by voxelwise subtraction of pre-gadolinium from post-gadolinium T1-weighted (T1w) images. The magnetization transfer ratio (MTR) image was derived voxelwise as $(M_0 - M_{SAT})/M_0$. Animal M#5.

The online version of this article includes the following figure supplement(s) for figure 6:

**Figure supplement 1.** Workflow of the magnetic resonance imaging (MRI) postprocessing pipeline.

**Figure supplement 2.** Postprocessed proton-density weighted images used as input for the convolutional neural network-based lesion segmentation.

Marmosets were usually housed with their twin counterpart, to maximize social interactions and enhance psychosocial wellbeing. Animals were weighed and monitored daily to ensure adequate nutritional intake and physical wellbeing. All protocols were approved by the National Institutes of Neurological Disorders and Stroke (NINDS) Institutional Animal Care and Use Committee (IACUC). Specifically, the neuroethics committee of the NINDS was consulted, and formally went through the protocol prior to submission on crucial topics including minimization of pain, justification of number of animals and the sex ratio, and dosing of methylprednisolone based on human data (IACUC protocol number #1308).

## Marmoset in vivo MRI

All six marmosets were scanned weekly under anesthesia, as previously described (*Sati et al., 2012*; *Lee et al., 2019*; *Lee et al., 2018*). We used a PDw sequence, which is sensitive to demyelination (*Reich, 2017*), to visualize lesions in vivo. T1w and MTR images were also obtained (*Figure 6*). T1w scans were repeated after injection of gadolinium-based contrast material (gadobutrol, 0.3 mmol/kg; triple the dose typically used in human clinical practice) to visualize enhancing lesions. Specific parameters for the different MRI sequences are listed in *Table 5*.

To minimize harm or pain in animals, all procedures, including intravenous access for gadolinium contrast material injection, were done under anesthesia. For post-anesthesia recovery, animals were

**Table 5.** Main parameters used for the different MRI contrasts acquired in vivo.

| MRI contrast | PDw | T1w | T2w | T2*w | MTR* |
|---|---|---|---|---|---|
| Sequence | 2D RARE | 2D MDEFT | 2D RARE | 2D MGE | 3D MGE |
| FOV (mm) | 32×24 | 32×24 | 32×24 | 32×24 | 38.4×38.4 |
| Matrix | 214×160 | 214×160 | 214×160 | 214×160 | 256×256 |
| Number of slices | 36 | 36 | 36 | 36 | 36 |
| Slice thickness (mm) | 1 | 1 | 1 | 1 | 1 |
| TR (ms) | 2300 | 12.5 | 8000 | 2150 | 20 |
| TE (ms) | 16 | 4.2 | 72 | 18 | 5 |
| TI (ms) | N/A | 1200 | N/A | N/A | N/A |
| ETL | 1 | N/A | 4 | N/A | N/A |
| Excitation pulse (shape, FA) | Sinc3, 90° | Sinc3, 12° | Sinc3, 90° | Sinc3, 70° | Sinc3, 10° |
| Refocusing pulse (shape, FA) | Sinc3, 180° | N/A | Sinc3, 180° | N/A | N/A |
| Preparation pulse (type, shape, FA, offset) | N/A | Excitation/inversion, Sech, 90°/180° | N/A | N/A | MT, Gauss, 1500 Hz |
| NEX | 1 | 1 | 2 | 1 | 2 |
| AT | 7 min 40 s | 6 min 56 s | 13 min 20 s | 7 min 10 s | 7 min 58 s |

FOV = field of view; TR = repetition tine; TE = echo time; TI = inversion time; ETL = echo train length (or RARE factor); FA = flip angle; NEX = number of repetitions; AT = acquisition time; RARE = rapid acquisition with relaxation and enhancement; MDEFT = modified drive equilibrium Fourier transform; MGE = multi-gradient echo. *Sequence was performed twice: with ($M_{SAT}$) and without ($M_0$) the MT pre-pulse.

gently woken with warm blankets and returned to their housing only after the animals were back to pre-anesthesia baseline, including spontaneous breathing, physical activity, and interactivity.

## In vivo MRI analysis of EAE lesions

Images were postprocessed using an in-house pipeline, which included an N3 intensity correction, image cropping, a multicontrast registration aiming to align every set of images on PDw sequence at the resolution of 150 µm in-plane (1 mm thickness), skull-stripping, and intensity normalization to gray matter signal intensity, which was set to 1, over the whole set of images (*Figure 6—figure supplement 1*).

Focal marmoset EAE WML were detected on PDw images using an automated convolutional neural network-based segmentation algorithm (*Roy et al., 2018b*; *Roy et al., 2018a*), and segmentations were subsequently verified and edited as needed. The network was trained using PDw MRI images from three EAE marmosets scanned in a previous study along with binary manual segmentation of lesions. An atlas consists of a pair of postprocessed PDw images (baseline and a timepoint) and the binary lesion segmentation of that timepoint. The baseline is assumed and was verified to be lesion-free. Once the network was trained, it was applied to the serial PDw images collected on all six animals (*Figure 6—figure supplement 2*), and lesion masks were automatically generated. Any lesion smaller than 4 voxels (0.0225 µL) was considered below the threshold and was removed from analysis.

For temporal progression computation, a lesion at timepoint $t=t_i$ was identified as the same lesion at timepoint $t=t_{i+1}$ if they overlapped by at least 4 voxels in 3D. All automated lesion segmentations were reviewed by an experienced rater. For subsequent MRI analysis of lesion trajectories, including intensity changes over time, all lesions were identified on every scan of each marmoset, and the average intensities were calculated using the automatically segmented lesions. For prelesion timepoints, a region of interest (ROI) of the size of the maximum lesion volume was drawn manually and centered over the white matter area where the lesion later appeared. When a lesion disappeared during the time course of the disease, an ROI of the maximum lesion size was propagated until the terminal scan.

## MRI WML categorization

MRI WML categorization and prediction were set based on our prior experience and performed independently by two experienced raters, blinded to histology. WML were classified according to the terminal MRI as follows: 1 – *predicted early active*, described as hyperintense on PDw and enhancing on T1w scans after injection of gadolinium; 2 – *predicted chronic, at least partially demyelinated*, described as hyperintense on PDw and not enhancing on T1w scans after injection of gadolinium; and 3 – *predicted remyelinated*, described as initially hyperintense on PDw, isointense on PDw at the terminal scan, and not enhancing on T1w scans after injection of gadolinium.

## Brain tissue preparation

Marmoset brains were collected immediately after death once the animals met the study endpoint. Brain tissue was processed using formalin fixation, paraffin embedding, and subsequent histopathological staining, as described previously (*Lee et al., 2019*; *Lee et al., 2018*; *Luciano et al., 2016*; *Absinta et al., 2014*). Briefly, an ultrahigh-resolution, ex vivo, 3D MRI of extracted brains was used to create individualized brain cradles with a 3D printer, which was in turn used to guide cutting of the brains into 2–4 mm slabs in an extremely close plane and axis to that of the in vivo MRI. Postmortem histological processing failed for animal M#1's brain.

## Histopathology of WML and NAWM

For visualizing myelin, Luxol fast blue (LFB) staining with periodic acid Schiff (PAS) counterstain and immunohistochemistry for myelin PLP were used. For characterizing inflammation and edema, hematoxylin and eosin (HE) and immunohistochemistry for ionized calcium-binding adaptor molecule (Iba1), CD3, and CD20, were used. Oligodendrocytes and OPC were assessed with aspartoacylase (ASPA) and oligodendrocyte transcription factor 2 (Olig2) double staining (mature oligodendrocytes are considered ASPA and Olig2 positive; OPC are considered Olig2 positive but ASPA negative). For axon staining, Bielschowsky's silver method was used. Briefly, deparaffinized slides were covered with 20% $AgNO_3$ and incubated at 40°C inside a dark chamber for 30 min. Slides were washed and placed in ammonia silver solution, prepared by adding concentrated ammonium hydroxide drop-by-drop into $AgNO_3$ until brown precipitate disappeared, at 40°C for 30 min. Developer working solution was added to the slides, made with developer stock solution (37–40% formaldehyde, citric acid, and nitric acid), ammonium hydroxide, and distilled water. After all incubations, slides were washed with 1% ammonium hydroxide, washed in distilled water, and treated with 5% sodium thiosulfate solution. Detailed immunohistochemical methods are provided in *Supplementary file 1*.

## Histological WML categorization

Histological analysis and characterization were performed by one experienced rater blinded to the MRI. The categorization was performed according to our experience with marmoset EAE lesions as detailed in previous publications (*Lee et al., 2019*; *Lee et al., 2018*; *Maggi et al., 2014*). The lesions were categorized as follows: 1 – *early active*: LFB-PAS shows prominent demyelination with LFB+, PAS+, and/or PLP+ phagocytes, indicating ingestion of myelin breakdown products (*Figure 1C*). PLP immunohistochemistry also demonstrates demyelination with myelin debris. ASPA/Olig2 double immunohistochemistry demonstrates loss of oligodendrocytes. Qualitative assessment highlights prominent Iba1[+] cell infiltration, CD3+ and CD20+ cells in the perivascular cuff and lesion core (not shown), and loss of axons on Bielschowsky silver staining. HE staining shows edema marked by irregular clear spaces around cells. 2 – *Chronic, likely at least partially demyelinated*: LFB-PAS staining and PLP immunohistochemistry show areas of complete demyelination (*Figure 2C*). Lesions contain few Iba1+ cells, and CD3+ and CD20+ cells (not shown) are scarce and only found around vessels. There is loss of both oligodendrocytes and OPC on ASPA/Olig2 staining. There is less edema compared to early active lesions (HE staining), and there is substantial loss of axons (Bielschowsky silver stain). 3 – *Remyelinated*; LFB-PAS staining and PLP immunohistochemistry show nearly normal myelin structure (*Figure 3C*). Both oligodendrocytes and OPC are present, as demonstrated by staining with ASPA/Olig2. Inflammatory cells are less prominent, with few infiltrations of Iba1+ microglia/macrophages or CD3+ and CD20+ lymphocytes (not shown). Bielschowsky staining shows some preservation of normal axon structures.

Quantitative measurement of demyelination and remyelination was performed by a single experienced rater (MD). To obtain a quantitative measurement of demyelination and remyelination, the percentage of demyelinated area for each lesion was extracted on LFB and PLP staining. For consistency, an ROI of 1500 $\mu m^2$ centered on each lesion core was placed; this ROI was large enough to include even the biggest lesion in our sample. The demyelinated area for each stain was calculated using the thresholding tool on FIJI (*Schindelin et al., 2012*) as follows: number of pixels with a null value*100/total number of pixels. ASPA+/Olig2+ cell count (oligodendrocytes) and ASPA-/Olig2+ cell count (OPC) were assessed using thresholding tools in FIJI45 within the same ROI for each lesion.

## Statistical analysis

To evaluate the statistical sensitivity and specificity of in vivo MRI detection of chronic demyelination or remyelination, relative to histopathology, we created confusion matrices and calculated true or false positive and negative rates, as well as sensitivity and specificity. For interrater reliability of MRI-predicted remyelination, we calculated Cohen's kappa. To test the effects of corticosteroid treatment and sex on remyelination, we used the point biserial correlation model.

## Acknowledgements

This study was funded by the National Institute of Neurological Disorders and Stroke Intramural Research Program and through a Cooperative Research and Development Agreement with Vertex Pharmaceuticals. We thank Dr. Tianxia Wu for help with statistical analyses. We thank everyone who took care of the animals and assisted with acquisition of MRI scans.

## Additional information

### Competing interests

Mac Johnson: is a shareholder and employee of Vertex Pharmaceuticals, Inc. The other authors declare that no competing interests exist.

### Funding

| Funder | Grant reference number | Author |
| --- | --- | --- |
| National Institute of Neurological Disorders and Stroke | Intramural Research Program | Maxime Donadieu Nathanael J Lee Seung-Kwon Ha Nicholas J Luciano Benjamin Ineichen Emily C Leibovitch Cecil C Yen Afonso C Silva Steve Jacobson Pascal Sati Daniel S Reich |
| Adelson Family Foundation | | Maxime Donadieu Seung-Kwon Ha Daniel S Reich |
| Vertex Pharmaceuticals | | Mac Johnson Daniel S Reich |
| Swiss National Science Foundation | | Benjamin Ineichen |
| National Multiple Sclerosis Society | RG-1907-34570 | Dzung L Pham Snehashis Roy |
| National Institutes of Health | R21OD030163 | Dzung L Pham Snehashis Roy |

The funders had no role in study design, data collection and interpretation, or the decision to submit the work for publication.

## Author contributions
Maxime Donadieu, Data curation, Formal analysis, Validation, Investigation, Methodology, Writing – original draft, Writing – review and editing; Nathanael J Lee, Conceptualization, Data curation, Formal analysis, Investigation, Methodology, Writing – original draft, Writing – review and editing; María I Gaitán, Writing – review and editing; Seung-Kwon Ha, Data curation, Formal analysis, Writing – original draft, Investigation, Methodology; Nicholas J Luciano, Data curation, Formal analysis, Investigation, Methodology; Snehashis Roy, Data curation, Formal analysis, Validation, Methodology; Benjamin Ineichen, Data curation, Formal analysis, Validation, Investigation, Methodology; Emily C Leibovitch, Data curation; Cecil C Yen, Dzung L Pham, Data curation, Formal analysis; Afonso C Silva, Supervision, Validation, Methodology; Mac Johnson, Supervision, Funding acquisition, Writing – review and editing; Steve Jacobson, Supervision, Project administration; Pascal Sati, Conceptualization, Data curation, Formal analysis, Supervision, Validation, Investigation, Methodology, Writing – original draft, Writing – review and editing; Daniel S Reich, Conceptualization, Resources, Funding acquisition, Validation, Investigation, Methodology, Writing – review and editing, Supervision

## Author ORCIDs
Maxime Donadieu ![ORCID] http://orcid.org/0009-0003-7349-1648
Steve Jacobson ![ORCID] http://orcid.org/0000-0003-3127-1287
Pascal Sati ![ORCID] http://orcid.org/0000-0002-6763-0125
Daniel S Reich ![ORCID] http://orcid.org/0000-0002-2628-4334

## Ethics
The study was performed under the guideline and in accordance with the National Institutes of Health IACUC. Specifically, the neuroethics committee of the National Institutes of Neurological Diseases and Stroke formally went through our manuscript prior to submission on salient topics including minimization of pain, justification of number of animals and the sex ratio, dosing of methylprednisone based on available human data. All procedures were performed under anesthesia to minimize discomfort and pain. Animals were housed in pairs or triplets to maximize social interactions and well-being. The institutional IACUC protocol number is #1308.

## Decision letter and Author response
Decision letter https://doi.org/10.7554/eLife.73786.sa1
Author response https://doi.org/10.7554/eLife.73786.sa2

# Additional files

## Supplementary files
• Supplementary file 1. Immunohistochemistry methodology. For each of the immunohistochemical targets, respective source, clonalities and hosts, and methods for antigen retrieval, blocking, and primary and secondary antibody inoculation are listed. HIER = heat-induced epitope retrieval; RT = room temperature; *P*=polyclonal antibody; M=monoclonal antibody; Rb = rabbit; Ms = mouse

• Transparent reporting form

• Source data 1. First part of serial in vivo magnetic resonance images of animal #1, including proton density-weighted sequence, used for analysis and figure generation. Uploaded into two separate source data for size of files. Files are in neuroimaging informatics technology initiative (NIFTI) format.

• Source data 2. Second part of serial in vivo magnetic resonance images of animal #1, including proton density-weighted sequence, used for analysis and figure generation. Uploaded into two separate source data for size of files. Files are in neuroimaging informatics technology initiative (NIFTI) format.

• Source data 3. First part of serial in vivo magnetic resonance images of animal #2, including proton density-weighted sequence, used for analysis and figure generation. Uploaded into two separate source data for size of files. Files are in neuroimaging informatics technology initiative (NIFTI) format.

• Source data 4. Second part of serial in vivo magnetic resonance images of animal #2, including proton density-weighted sequence, used for analysis and figure generation. Uploaded into two separate source data for size of files. Files are in neuroimaging informatics technology initiative (NIFTI) format.

• Source data 5. Serial in vivo magnetic resonance images of animal #3, including proton density-

weighted sequence, used for analysis and figure generation. Files are in neuroimaging informatics technology initiative (NIFTI) format.

• Source data 6. Serial in vivo magnetic resonance images of animal #4, including proton density-weighted sequence, used for analysis and figure generation. Files are in neuroimaging informatics technology initiative (NIFTI) format.

• Source data 7. Serial in vivo magnetic resonance images of animal #5, including proton density-weighted sequence, used for analysis and figure generation. Files are in neuroimaging informatics technology initiative (NIFTI) format.

• Source data 8. Serial in vivo magnetic resonance images of animal #6, including proton density-weighted sequence, used for analysis and figure generation. Files are in neuroimaging informatics technology initiative (NIFTI) format.

• Source data 9. Iba1 and proteolipid protein (PLP) immunohistochemistry images of animals #2–6 used for analysis and figure generation.

### Data availability

All of the 6 marmosets' serial in vivo MRI images, including all the sequences used for analysis and figure generation, were uploaded in an easily accessible format (NIFTI). The file names are titled with the corresponding animal # used in the manuscript, as well as the date of MRI acquisition. All the Iba1 and PLP immunohistochemistry stains have been uploaded as well.

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
