## [Editor Report]

This important study combined MRI(magnetic resonance imaging) imaging and histopathology to examine the remyelination of brain lesions in an EAE marmoset model of multiple sclerosis. This work addresses in a non-human primate a missing link in the neuropathology of myelin repair, because in human MS it is virtually impossible to study the lesion dynamics by MRI (in live patients) and demyelination by histology (upon brain autopsy). The data presented are solid although the conclusions would have been stronger with further histological evidence of remyelination.

---

## [Decision Letter]

**Decision letter after peer review:**

Thank you for submitting your article "in vivo MRI of Endogenous Remyelination in a Nonhuman Primate Model of Multiple Sclerosis" for consideration by *eLife*. We appreciate your patience during the handling of this submission. Your article has been reviewed by 3 peer reviewers, and the evaluation has been overseen by a Reviewing Editor and Jeannie Chin as the Senior Editor. The reviewers have opted to remain anonymous.

The reviewers have discussed their reviews with one another, and the Reviewing Editor has drafted this response to help you prepare a revised submission. As you will see below, there was an agreement that your work is an important step towards the MRI-based diagnosis of CNS remyelination in multiple sclerosis. However, it was also felt that the present manuscript has shortcomings in the presentation of histological data and some missing information about data acquisition. Please look for details below how to improve a revised version of your manuscript that better supports the claims made.

*Reviewer #1 (Recommendations for the authors):*

Similar to the serial MRI, the manuscript would be enhanced by supplementing Iba1- and PLP-stained sections of each lesion.

*Reviewer #2 (Recommendations for the authors):*

Before publication I think that this would be improved by consideration of the following

How do the authors know that PDw is imaging new myelin formation rather than loss of inflammation? The signal is maintained in acute and chronic inflammation, and lost in remyelination -which is also associated with reduced inflammation. It's not clear to me how the two events can be distinguished by this imaging protocol. Is the time course of inflammatory loss different to that for every myelination? The gadolinium experiments mentioned in the methods but not included in the results might address this?

How is the distinction between acute and chronic inflammatory lesions made. Is it just the time of onset after immunisation?

In figure 5 it is not clear how the transition from demyelination to remyelination was established, and so the evaluation of the time course for remyelination is not clear

*Reviewer #3 (Recommendations for the authors):*

If possible, it would be useful to have images of lesions (histology) at somewhat lower magnification as well, to appreciate differences in density etc. This would be very helpful particularly in the case of Luxol Fast Blue staining of remyelinated lesions so that pale staining compared to non-affected tissue can be appreciated, and for ASPA/Olig2 and Bielschowsky staining, to illustrate differences in density.

Regarding Iba1 staining, clear cellular labeling pattern is observed only in Figure 4, while Figure 2 shows high background and very few if any clearly labeled cells, which is a bit better in Figure 3.

Points to clarify:

1. Figure 5 shows the estimated timing of demyelination/remyelination. Three out of four graphs are from methylprednisolone (MP)-treated animals. It was indicated in Methods that MP administration was performed after demyelination was first detected, but the first graph shows 1 double bar, at 12 weeks (prior to the first peak), and a triple bar at 18 weeks approx. Was this animal treated twice?

2. In the same figure, 2 lesions from M#5 are shown for which, as the legend indicates, remyelination was confirmed histologically. Yet, on page 16, it is indicated that remyelination was detected in animals M#2-4 by MRI and confirmed by histology, and that M#5 had a single remyelinated lesion not seen on MRI, which seems to contradict Figure 5. Could this be clarified?

[Editors' note: further revisions were suggested prior to acceptance, as described below.]

Thank you for resubmitting your work entitled "in vivo MRI of Endogenous Remyelination in a Nonhuman Primate Model of Multiple Sclerosis" for further consideration by *eLife*. Your revised article has been evaluated by Jeannie Chin (Senior Editor) and a Reviewing Editor.

The manuscript has been improved but there are remaining issues that need to be addressed, as described in the individual reviewer comments at the end of this correspondence. A brief summary of those essential issues follows:

Essential revisions:

1) Related to the finding that none of the lesions returning to isointensity on PDw MRI demonstrated gadolinium enhancement on T1w scans, while the gadolinium data is very helpful, no quantification is provided. This should be added. Ideally, additional histological evidence of remyelination should be provided to strengthen the power of PDw to classify the lesions. Also, please clarify whether gadolinium enhancement is also considered as a marker of acute versus chronic lesions in this paper.

2) The authors assert that prior data shows that the changes seen occur at the same time as remyelination and don't correlate with any changes in inflammation, but this was not sufficiently addressed. Please include a more detailed analysis of remyelination and inflammation in the lesions on which the MRI was performed, rather than a comparison with older data. Of particular interest would be histological evidence of remyelination such as reduced myelin density, shorter and thinner myelin internodes and myelin protein staining of remyelinating oligodendrocyte cytoplasm.

3) A concern raised in the previous review was that it was not possible to see loss of oligodendroglia based on the image provided to illustrate acute lesions (now in Figure 1), in which there is a small oligo-free area surrounding what may be blood vessels, but the remaining PLP-free area contains many oligodendroglial cells. The question of oligodendroglial loss could be addressed by quantifying the density in perilesional area and comparing it to the intralesional density. However, the authors simply refer to a previous manuscript (JCI, 2019). Were the same animals/lesions analyzed in this study? This point needs to be clearly addressed throughout the manuscript wherever references to previous data are made. Please add quantitative data to demonstrate oligo depletion followed by repopulation in remyelinated lesion. If the quantification does not support the loss of oligodendroglia, then these claims should be removed.

4) The histological criteria for lesion classification is slightly confusing, which was brought up in the original review as well. The authors assert that experienced histopathologists blindly classified these lesions in Kuhlmann 2017, but it is not really clear that exactly the same immunolabelings were performed. Please provide a clear statement on which histological parameters (immunolabelings) were considered by the histopathologists to classify the lesions in this study.

Note: Because the results of this manuscript have important implications for pre-clinical testing of remyelinating drugs, solid data on the lesions analyzed in this manuscript should be provided to reinforce the conclusions.

*Reviewer #1 (Recommendations for the authors):*

There is no histological evidence of remyelination such as reduced myelin density, shorter and thinner myelin internodes and myelin protein staining of remyelinating oligodendrocyte cytoplasm. Most lesional areas are small. It is most likely that many of the MRI ROIs are due to alterations in myelinated white matter. It may be that all lesion are remyelinated to a point that they cannot be distinguished from normal myelinated white matter. It would seem unlikely that a lesion in the process of remyelination is not present in all the lesions investigated. This reduces the utility of the model to investigate remyelination. What can be measured? Unlikely that remyelination can be enhanced. I do not consider this as a reliable model for remyelination. This a relevant question based upon the species utilized in this study.

*Reviewer #2 (Recommendations for the authors):*

My key point of concern was

"How do the authors know that PDw is imaging new myelin formation rather than loss of inflammation? The signal is maintained in acute and chronic inflammation, and lost in remyelination -which is also associated with reduced inflammation. It's not clear to me how the two events can be distinguished by this imaging protocol. Is the time course of inflammatory loss different to that for every myelination? The gadolinium experiments mentioned in the methods but not included in the results might address this?"

This is rebutted as follows

"- This is an essential point, and we thank the reviewer for the inquiry. We agree (and have now clarified in the paper) that PDw imaging is sensitive to inflammation and other changes; the lack of specificity to myelin is indeed well known in the field. What is critical for our purposes, however, is the ability to infer changes in myelin from the combination of in-vivo MRI data and extensive knowledge of the time course of lesion development and repair, which in our hands is derived from the analysis of ~100 animals over more than a decade, as reflected in our prior publications and the current manuscript. Hence, our paper is essentially about sensitivity of PDw MRI to myelin changes in this context, not about specificity of the technique. It is worth noting, in this context, that chronic EAE lesions are typically either gliotic (chronic inactive) or remyelinated, with few inflammatory cells; thus, the difference in MRI signal between remyelinated and chronically demyelinated lesions cannot be explained purely by inflammation. Furthermore, none of the lesions returning to isointensity on PDw MRI demonstrated gadolinium enhancement on T1w scans. In the revision, we have added material under the description on remyelinated lesions in the Results section."

The gadolinium data is very helpful, although no quantification is provided. This should be added.

I take the point that the prior data argues that the changes seen occur at the same time as remyelination and don't correlate with any changes in inflammation, but feel my concern is not really addressed in the discussion despite it being recognised as "essential". At the very least I'd like to see an addition to the discussion here, although Id much prefer a more detailed analysis of remyelination and inflammation in the lesions on which the MRI was performed rather than a comparison with older data

*Reviewer #3 (Recommendations for the authors):*

My general feeling is that, while several issues were successfully addressed by this revision, referring to previous manuscripts has not provided answers to some fundamental questions raised in the previous review.

One of my points was that I could not see loss of oligodendroglia based on the image provided to illustrate acute lesions (now in Figure 1), in which there is a small oligo-free area surrounding what may be blood vessels, but the remaining PLP-free area contains many oligodendroglial cells. The question of oligodendroglial loss could be addressed by quantifying the density in perilesional area and comparing it to the intralesional density. However, the authors simply refer to a previous manuscript (JCI, 2019). Were the same animals/lesions analyzed in this study? This is really not clear from the manuscript. One sentence in the Discussion refers to previous analyses of oligo repopulation and remyelination, but is it the same lesions/animals? This should be clarified.

Because some acutely demyelinating lesions show oligo preservation, I do not think that showing a decrease in oligos is necessary to claim that the lesion is demyelinated, but I do think that if this claim is made, it should be supported by data, especially when the image provided does not illustrate the claim. Otherwise, the claim should be omitted. Thus, I recommend adding some quantitative data to demonstrate oligo depletion followed by repopulation in remyelinated lesions. If no depletion is evidenced in acute lesions, then it can be discussed that demyelination with oligodendroglial preservation has also been observed in early MS lesions. If these data on the same lesions are provided in previous publications, it should be clearly indicated that these are the same lesions.

A crucial issue raised in the previous round was that of PDw sequences detecting changes other than demyelination/remyelination. The authors did mention that gadolinium enhancement was present in acute but not chronic lesions, both of which were demyelinated and show similar Pdw, which, in my opinion, does suggest that changes in PDw signal are not due to inflammatory infiltrates, even though does not exclude that other events (axonal changes, astrocytes, diffuse microgliosis) might alter the signal. Including gadolinium data and potentially, quantification, would be a good argument to exclude inflammatory infiltrates as the cause of changes in PDw. This point should be reinforced in the Discussion. Ideally, additional histological evidence of remyelination should be provided to strengthen the power of PDw to classify the lesions.

I still find the histological criteria for lesion classification slightly confusing. The answer to my suggestion to clearly define lesion classification criteria applied in this paper was that experienced histopathologists blindly classified these lesions based on Tanja Kuhlmann paper from 2017, but from the data presented, it is not really clear that exactly the same immunolabelings were performed (as compared to Kuhlmann paper). From the authors' response, it can be concluded that all these analyses were performed in their previous work on marmoset lesions (again, were these the same lesions?), and thus not repeated in this study. I think that the authors need to provide a clear statement on which histological parameters (immunolabellings) were considered by the histopathologists to classify the lesions in this study.

The authors state that lesions were classified as acute if younger than 5 weeks and remyelinated or chronic if older than 5 weeks. I suppose this refers to MRI classification of the lesions, and not the histological one. As the authors also describe acute lesions as gadolinium enhancing, and chronic lesions (and remyelinated ones) gadolinium enhancement-free, was gadolinium enhancement also considered as a marker of acute versus chronic lesions in this paper?

Because the results of this manuscript have important implications for pre-clinical testing of remyelinating drugs, solid data on the lesions analyzed in this manuscript should be provided to reinforce the conclusions.

---

## [Author Response]

Reviewer #1 (Recommendations for the authors):Similar to the serial MRI, the manuscript would be enhanced by supplementing Iba1- and PLP-stained sections of each lesion.

We agree that supplementing all the lesions with Iba1 and PLP stains would be a useful addition and have now added them to the supplementary data section of the manuscript, under Source data 9.

Reviewer #2 (Recommendations for the authors):Before publication I think that this would be improved by consideration of the followingHow do the authors know that PDw is imaging new myelin formation rather than loss of inflammation? The signal is maintained in acute and chronic inflammation, and lost in remyelination -which is also associated with reduced inflammation. It's not clear to me how the two events can be distinguished by this imaging protocol. Is the time course of inflammatory loss different to that for every myelination? The gadolinium experiments mentioned in the methods but not included in the results might address this?

This is an essential point, and we thank the reviewer for the inquiry. We agree (and have now clarified in the paper) that PDw imaging is sensitive to inflammation and other changes; the lack of specificity to myelin is indeed well known in the field. What is critical for our purposes, however, is the ability to infer changes in myelin from the combination of in-vivo MRI data and extensive knowledge of the time course of lesion development and repair, which in our hands is derived from the analysis of ~100 animals over more than a decade, as reflected in our prior publications and the current manuscript. Hence, our paper is essentially about sensitivity of PDw MRI to myelin changes in this context, not about specificity of the technique.

It is worth noting, in this context, that chronic EAE lesions are typically either gliotic (chronic inactive) or remyelinated, with few inflammatory cells; thus, the difference in MRI signal between remyelinated and chronically demyelinated lesions cannot be explained purely by inflammation. Furthermore, none of the lesions returning to isointensity on PDw MRI demonstrated gadolinium enhancement on T1w scans. In the revision, we have added material under the description on remyelinated lesions in the Results section.

How is the distinction between acute and chronic inflammatory lesions made. Is it just the time of onset after immunisation?

We thank the reviewer for the opportunity to clarify this. The differentiation between acute and chronic is solely based on the timeline of the lesion progression, detected by the serial in vivo MRI, and not on the time after immunization.

In figure 5 it is not clear how the transition from demyelination to remyelination was established, and so the evaluation of the time course for remyelination is not clear

We thank the reviewer for pointing out the unclear time course as highlighted on Figure 5 (now Figure 4). Our evaluation was based on the MRI signal intensity change. We have clarified this point in our manuscript in the figure legend.

Reviewer #3 (Recommendations for the authors):If possible, it would be useful to have images of lesions (histology) at somewhat lower magnification as well, to appreciate differences in density etc. This would be very helpful particularly in the case of Luxol Fast Blue staining of remyelinated lesions so that pale staining compared to non-affected tissue can be appreciated, and for ASPA/Olig2 and Bielschowsky staining, to illustrate differences in density.

As mentioned above, we agree that lower magnification images will be helpful to identify areas of thinly remyelinated regions and have included these.

Regarding Iba1 staining, clear cellular labeling pattern is observed only in Figure 4, while Figure 2 shows high background and very few if any clearly labeled cells, which is a bit better in Figure 3.

We included both a higher and lower magnification of the Iba1 staining on Figure 2 (now Figure 1) for better visualization. We hope this clarifies the issue.

Points to clarify:1. Figure 5 shows the estimated timing of demyelination/remyelination. Three out of four graphs are from methylprednisolone (MP)-treated animals. It was indicated in Methods that MP administration was performed after demyelination was first detected, but the first graph shows 1 double bar, at 12 weeks (prior to the first peak), and a triple bar at 18 weeks approx. Was this animal treated twice?

We apologize for the lack of clarity in our description. As you mentioned, the animal was treated twice after the first steroid treatment did not result in any clinical or radiological improvement. We have clarified this in our figure (now Figure 4) legend.

2. In the same figure, 2 lesions from M#5 are shown for which, as the legend indicates, remyelination was confirmed histologically. Yet, on page 16, it is indicated that remyelination was detected in animals M#2-4 by MRI and confirmed by histology, and that M#5 had a single remyelinated lesion not seen on MRI, which seems to contradict Figure 5. Could this be clarified?

We thank the reviewer for catching this error; we have now clarified and corrected the error in our revised manuscript.

[Editors' note: further revisions were suggested prior to acceptance, as described below.]

Essential revisions:1) Related to the finding that none of the lesions returning to isointensity on PDw MRI demonstrated gadolinium enhancement on T1w scans, while the gadolinium data is very helpful, no quantification is provided. This should be added. Ideally, additional histological evidence of remyelination should be provided to strengthen the power of PDw to classify the lesions. Also, please clarify whether gadolinium enhancement is also considered as a marker of acute versus chronic lesions in this paper.

We appreciate the comment and have made substantial changes to the paper in response.

First, we would like to clarify that most of the lesions newly detected on PDw MRI showed gadolinium enhancement on T1w images. Only lesions identified as “early active” on pathology were found to be enhancing on the terminal MRI. We added the following paragraph in the Results section:

“T1w gadolinium enhancement as a marker of acute inflammation

Across the 6 animals scanned longitudinally, 82% of the lesions newly detected on PDw MRI presented T1w gadolinium enhancement. Enhancement was occasionally seen at the following timepoint (10–15 days after first detection). No chronic, at least partially demyelinated or remyelinated WML presented gadolinium enhancement on the terminal scan. M#5–6 presented at least one early active WML enhancing lesion at their terminal scan.”

The use of gadolinium enhancement to categorize lesions as acute or chronic has now been clarified in the Results section, in the section entitled “MRI WML categorization.”

Second, as suggested, we have added more histological evidence of lesion remyelination in the revision. To do so, we assessed, over 31 lesions, the percentage of unstained area on PLP and LFB stains for myelin protein and lipid, respectively. We also added 10 NAWM areas from the same animals as control. As expected, we observed a larger unstained area for demyelinated lesions vs remyelinated lesions on both PLP and LFB. We have added details to the method section as well as a detailed paragraph in results:

“Histological quantification recapitulates MRI rater analysis of lesion myelin status

Assessment of PLP staining in 31 lesions and 10 normal appearing white matter (NAWM) areas (1500 µm^2^ centered over the lesion core) demonstrated larger unstained areas in early active (58±25%) and chronic, at least partially demyelinated (38±25%) lesions compared to remyelinated lesions (4.5±1.1%) or NAWM (2.6±0.3%) (Figure 5A). LFB assessment showed similar results: 63±26% unstained area in early active lesions, 43±26% in chronic, at least partially demyelinated lesions, 15±25% in remyelinated lesions, and 3.0±0.4% in NAWM (Figure 5B). We observed a significantly smaller unstained PLP area in NAWM compared to remyelinated lesions (2-sample t-test, p<0.001). There were no apparent differences between PLP and LFB staining in the different lesion categories and in NAWM (2-sample t-test). Interestingly, 1 lesion appeared remyelinated on PLP, with less than 6% of unstained, area but demyelinated on LFB (82% of unstained area).”

Oligodendrocyte density was addressed with ASPA and Olig2 staining. ASPA+/Olig2+ cells, reflecting mature oligodendrocytes, and ASPA-/Olig2+ cells, potentially reflecting OPC, were counted for every lesion in comparison to 10 NAWM areas. Our analysis showed a higher ASPA+/Olig2+ cell count in remyelinated lesions compared to acute or chronic lesions as well as negative correlation between lesion volume at terminal PDw MRI and ASPA+/Olig2+ cell number. A detailed paragraph was added in the Results section:

“Oligodendrocyte and OPC counts are consistent with degree of demyelination in lesions Quantitative assessment of ASPA+/Olig2+ (mature oligodendrocytes) and ASPA-/Olig2+ (OPC) across the 31 lesions and 10 NAWM areas showed, as expected, more mature oligodendrocytes in remyelinated lesions and NAWM than early active or chronic, at least partially demyelinated lesions (Figure 5C). Consistent with PLP observations, we observed significantly more oligodendrocytes in NAWM compared to remyelinated lesions (2-sample t-test, p<0.001). Interestingly, more OPC were found in early active demyelinating lesions than in remyelinated lesions or NAWM (Figure 5D). Younger lesions (<10 weeks old by MRI) had more OPC than older lesions, highlighted by a negative correlation between lesion age and OPC count (r = -0.46; p = 0.009).”

The revision includes a newly added Figure 5, which summarizes the quantitative histological analysis. Adequate description in the method section was also added (“Histological WML categorization”). We have also added additional comments in the Discussion section.

2) The authors assert that prior data shows that the changes seen occur at the same time as remyelination and don't correlate with any changes in inflammation, but this was not sufficiently addressed. Please include a more detailed analysis of remyelination and inflammation in the lesions on which the MRI was performed, rather than a comparison with older data. Of particular interest would be histological evidence of remyelination such as reduced myelin density, shorter and thinner myelin internodes and myelin protein staining of remyelinating oligodendrocyte cytoplasm.

As requested and described in the response to essential revision #1 (above), we have added additional histological analysis to back up and quantify our findings regarding remyelination.

In addition to myelin staining, we did additional characterization of CD3+ T cells and CD20+ B cells to assess inflammation in each type of lesion. More details were added in the paragraph “Histological WML categorization” in the method section for each lesion category.

We also included enhanced discussion on those points in the appropriate section of the paper.

We added acknowledgement in the discussion that additional immunohistochemistry would be useful to further characterize myelination status, but unfortunately we were not able to achieve this in our tissue:

“Finally, we did not obtain ultrathin sections to quantify myelin thickness, as we were interested in performing a battery of stains on our tissue, and as lesions were found at a variety of orientations relative to axons, which would complicate such quantification.”

3) A concern raised in the previous review was that it was not possible to see loss of oligodendroglia based on the image provided to illustrate acute lesions (now in Figure 1), in which there is a small oligo-free area surrounding what may be blood vessels, but the remaining PLP-free area contains many oligodendroglial cells. The question of oligodendroglial loss could be addressed by quantifying the density in perilesional area and comparing it to the intralesional density. However, the authors simply refer to a previous manuscript (JCI, 2019). Were the same animals/lesions analyzed in this study? This point needs to be clearly addressed throughout the manuscript wherever references to previous data are made. Please add quantitative data to demonstrate oligo depletion followed by repopulation in remyelinated lesion. If the quantification does not support the loss of oligodendroglia, then these claims should be removed.

We appreciate the request for clarification. With respect to the second point, the lesions and animals studied in this project were different those reported in our previous publication (JCI, 2019).

Please see the response to essential revision #1 (above) for the revisions we made to quantify OPC and oligodendrocytes. The new analysis is summarized in the newly added Figure 5, and the methods are described under “Histological WML categorization.”

4) The histological criteria for lesion classification is slightly confusing, which was brought up in the original review as well. The authors assert that experienced histopathologists blindly classified these lesions in Kuhlmann 2017, but it is not really clear that exactly the same immunolabelings were performed. Please provide a clear statement on which histological parameters (immunolabelings) were considered by the histopathologists to classify the lesions in this study.

In response to this critique, we reshaped the method section to give more detail on our lesion categorization for both MRI and histopathology. The following paragraphs were added or changed:

“MRI WML categorization

MRI WML categorization and prediction were set based on our prior experience and performed independently by 2 experienced raters, blinded to histology. WML were classified according to the terminal MRI as follows: 1 — predicted early active, described as hyperintense on PDw and enhancing on T1w scans after injection of gadolinium; 2 — predicted chronic, at least partially demyelinated, described as hyperintense on PDw and not enhancing on T1w scans after injection of gadolinium; and 3 — predicted remyelinated, described as initially hyperintense on PDw, iso-intense on PDw at the terminal scan, and not enhancing on T1w scans after injection of gadolinium.”

“Histological WML categorization

Histological analysis and characterization were performed by 1 experienced rater blinded to the MRI. The categorization was performed according to our experience with marmoset EAE lesions as detailed in previous publications^14,24,44^. The lesions were categorized as follows: 1 — early active: LFB-PAS shows prominent demyelination with LFB+, PAS+, and/or PLP+ phagocytes, indicating ingestion of myelin breakdown products (Figure 1C). PLP immunohistochemistry also demonstrates demyelination with myelin debris. ASPA/Olig2 double immunohistochemistry demonstrates loss of oligodendrocytes. Qualitative assessment highlights prominent Iba1+ cell infiltration, CD3+, and CD20+ cells in the perivascular cuff and lesion core (not shown), and loss of axons on Bielschowsky silver staining. HE staining shows edema marked by irregular clear spaces around cells. 2 — Chronic, likely at least partially demyelinated: LFB-PAS staining and PLP immunohistochemistry show areas of complete demyelination (Figure 2C). Lesions contain few Iba1+ cells, and CD3+ and CD20+ cells (not shown) are scarce and only found around vessels. There is loss of both oligodendrocytes and OPC on ASPA/Olig2 staining. There is less edema compared to early active lesions (HE staining), and there is substantial loss of axons (Bielschowsky silver stain). 3 — Remyelinated; LFB-PAS staining and PLP immunohistochemistry show nearly normal myelin structure (Figure 3C). Both oligodendrocytes and OPC are present, as demonstrated by staining with ASPA/Olig2. Inflammatory cells are less prominent, with few Iba1+ microglia/macrophages or CD3+ and CD20+ lymphocytes (not shown). Bielschowsky staining shows some preservation of normal axon structures.

Quantitative measurement of demyelination and remyelination was performed by a single experienced rater (MD). To obtain a quantitative measurement of demyelination and remyelination, the percentage of demyelinated area for each lesion was extracted on LFB and PLP staining. For consistency, a region of interest (ROI) of 1500 µm^2^ centered on each lesion core was placed; this ROI was large enough to include even the biggest lesion in our sample. The demyelinated area for each stain was calculated using the thresholding tool on FIJI45 as follows: number of pixels with a null value*100/total number of pixels. ASPA+/Olig2+ cell count (oligodendrocytes) and ASPA-/Olig2+ cell count (OPC) were assessed using thresholding tools in FIJI45 within the same ROI for each lesion.”